# Genome-Wide Identification of *WOX* Genes in Korean Pine and Analysis of Expression Patterns and Properties of Transcription Factors

**DOI:** 10.3390/biology14040411

**Published:** 2025-04-12

**Authors:** Qun Zhang, Xiuyue Xu, Ling Yang

**Affiliations:** 1State Key Laboratory of Tree Genetics and Breeding, Northeast Forestry University, Harbin 150040, China; zqzq19960420@163.com (Q.Z.); 15663593162@163.com (X.X.); 2College of Forestry, Beijing Forestry University, Beijing 100091, China

**Keywords:** bioinformatics, gene expression, Korean pine, protein interaction, WOX

## Abstract

In this study, the *WOX* gene family of Korean pine was analyzed, and 21 family members were identified using bioinformatics methods. The results showed that the total number of amino acids in all proteins is between 164 and 488 aa, with molecular weights between 18,456.30 and 53,893.18 Da and isoelectric point values between 5.56 and 9.80. According to the predicted results, the average hydrophilicity coefficient of the *PkWOX* proteins is between −0.993 and −0.565. The phylogenetic tree divides the *PkWOX* genes into three sub-branches. A total of 21 *PkWOX* genes are unevenly distributed on 7 of 12 chromosomes. The *PkWOX* promoters contain cis-acting elements such as stress response, light response, hormone response, and meristem regulation. GO entry for *PkWOX* is a biological process and molecular function. *PkWOX16* was expressed in all tissues. *PkWOX2*, *3* had higher expression in the embryonic callus, non-embryonic callus, somatic embryo, and zygotic embryo. *PkWOX2*, *3,* and *16* were located in the nucleus and in the cell membrane. The *PkWOX2* and *3* proteins exhibited transcriptional self-activation activity, while *PkWOX16* did not. In this study, the members of the WOX transcription factor family were identified and systematically analyzed in Korean pine, laying a foundation for the research on their functions.

## 1. Introduction

The WUSCHEL (WUS) transcription factor, belonging to the homeobox family, represents the archetype of the WUS-related homeobox (WOX) protein group, which is unique to plants and forms part of the broader homeodomain (HB) transcription factor superfamily. Generally, the homeodomain consists of 60 amino acids arranged in a helix–loop–helix–turn–helix conformation [1]. Some WOX family members have been found to possess non-canonical homeodomains with extended amino acid sequences [2]. This domain is highly conserved, can be encoded by specific DNA sequences, and regulates the expression of target genes at precise time points.

With the development of molecular technology, the functions of members of the *WOX* gene family have gradually been elucidated, and the functional research in the model plant *Arabidopsis thalian* is relatively clear. This has laid a solid research foundation for subsequent researchers to study *WOX* genes. In 2004, 14 homologous domains similar to those resembling WUS were identified in *A. thalian* and labeled as *WOX* genes [3]. Plant WOX proteins can be phylogenetically classified into three distinct evolutionary groups: the WUS lineage, the intermediate lineage, and the ancient lineage [4]. The members of the different branches show universal differences, and members of the same branch are conservative. The members of the modern branches promote stable differentiation of the stem tip meristems [5,6]. Proteins belonging to the intermediate clade primarily function in controlling zygote formation and initial embryogenesis [7]. Proteins within the ancient clade primarily modulate root-system formation and growth [8]. In recent years, the *WOX* gene has been discovered in more plants, tobacco (*Nicotiana tabacum*) [9], poplar (*Populus* sp.) [10], tomato (*Solanum lycopersicum*) [9], cotton (*Gossypium* spp.) [11], grape (*Vitis vinifera*) [12], *Liriodendron hybrids* [13], rice (*Oryza sativa*) [10], wheat (*Triticum aestivum*) [14], maize (*Zea mays*) [10], pine (*Pinus linn*) [15], and Norway spruce(*Picea abies*) [16].

Recent research has increasingly demonstrated the critical roles of WOX transcription factors in multiple biological processes, including organ regeneration, developmental regulation, stress adaptation, and transcriptional control, with particularly significant functions during somatic embryogenesis [17,18,19,20,21]. In maize (*Zea mays*), the *ZmWOX2A* gene, which shares homology with *A. thalian* WOX2, exhibits specific expression patterns during early embryogenesis. Its transcripts are initially observed in the zygote and persist through the multicellular stage, showing restricted localization to the embryonic apical domain [22]. Studies have revealed that *A. thalian* WUSCHEL (AtWUS) exhibits transcriptional activity as early as the 16-cell proembryo stage (Nic stage), where it plays a crucial regulatory role in somatic embryogenesis progression and subsequent embryo maturation processes [23]. The establishment of embryonic polarity in Arabidopsis involves the coordinated action of multiple WOX transcription factors, including *AtWOX2*, *AtWOX8*, and *AtWOX9*. These regulators exhibit distinct spatial expression patterns during early embryogenesis: AtWOX2 is predominantly localized to apical cell lineages, while AtWOX8 and AtWOX9 show basal cell-specific expression. This complementary expression pattern is critical for proper apical–basal axis formation during embryo patterning [3,4,5]. Furthermore, transcriptomic analyses have identified *PaWOX2* as a prominently expressed gene in the embryogenic tissues of Norway spruce (*Picea abies*) [16]; moreover, functional studies demonstrate that *PpWOX2* overexpression significantly alters morphological characteristics during somatic embryogenesis in maritime pine (*Pinus pinaster*) [21]. A recent genome-wide analysis revealed the presence of 11 WOX family genes in Masson pine (*Pinus massoniana*), with *PmWOX2*, *PmWOX3*, and *PmWOX4* exhibiting particularly abundant transcripts in callus tissues [24].

The Korean pine, as the dominant and foundational species of the broad-leaved Korean pine forest, holds significant ecological value [25]. Its timber and nuts have significant economic value and are extremely important high-value timber and nut species in China and East Asia. In the study of somatic embryogenesis of Korean pine, it was found that the transition from embryonic callus multiplication to somatic embryo maturation is one of the most difficult and critical steps. At present, there are serious problems with genotype dependency and loss of embryogenesis. Cell lines with the capacity for somatic embryogenesis gradually decrease their somatic embryogenic potential after long-term proliferation and culture until they completely lose their somatic embryogenesis and have a low yield of somatic embryos.

Through genome-wide screening of Korean pine, we characterized 21 *PkWOX* family members. Our integrated approach combined bioinformatics and transcriptomic analyses to investigate various molecular features, including (1) conserved protein domains and physicochemical parameters, (2) genomic distribution across chromosomes, (3) phylogenetic relationships and gene architecture, (4) regulatory cis-elements in promoter regions, (5) functional annotation through GO analysis, and (6) expression profiles in different tissues and under abiotic stress. Furthermore, three *PkWOX* members exhibiting elevated expression in callus tissues were selected for functional characterization, including subcellular localization and transactivation potential assays. These findings establish a solid theoretical framework for elucidating *WOX* gene functions during somatic embryogenesis in Korean pine. Notably, this study represents the first comprehensive genome-wide characterization of *WOX* family genes in this conifer species, including systematic analysis of their tissue-specific expression profiles. Importantly, the WUS gene, known for its crucial role in maintaining shoot apical meristem stem cells, was also identified and characterized in this investigation [26]. Studies in *A. thalian* have demonstrated that auxin-mediated induction of WUSCHEL (WUS) expression plays an essential role in maintaining pluripotency and promoting continuous proliferation of embryonic stem cells throughout somatic embryogenesis [27]. During cellular reprogramming processes, targeted upregulation of key embryonic patterning regulators *WOX2* and *WOX3* has been shown to significantly improve both the efficiency of somatic embryo formation and subsequent developmental progression [28]. Through integrated transcriptomic and bioinformatic analyses, we systematically investigated the expression profiles of *WOX* genes during physiological embryogenesis in Korean pine (*Pinus koraiensis*). Our findings provide essential genomic resources for elucidating the functional mechanisms of *WOX* genes in conifer embryogenesis. Given the critical roles of WOX transcription factors in plant development, a comprehensive genome-wide characterization of the *WOX* gene family in Korean pine—including identification of all family members and their spatiotemporal expression patterns—represents both an immediate research priority and a fundamental requirement for advancing our understanding of conifer embryogenesis.

## 2. Materials and Methods

### 2.1. Plant Material and Growth Conditions

The experimental material was the embryogenic callus of Korean pine stored in our laboratory, which consisted of MLV (Coolaber, Beijing, China) supplemented with 0.5 mg∙L^−1^ 2,4-D (aladdin, Shanghai, China), 0.1 mg∙L^−1^ 6-BA (Biotopped, Beijing, China), 30 g∙L^−1^ sucrose (Kermel, Tianjin, China), 0.5 g∙L^−1^ L-glutamine (Biotopped, Beijing, China), and 0.5 g∙L^−1^ CH (Biotopped, Beijing, China). The culture medium was adjusted to pH 5.8 before autoclaving. Subculturing was performed every 14 days under dark conditions at 24 ± 1 °C to sustain callus proliferation. For comparative studies, *Nicotiana tabacum* plants were cultivated in a 1:1 (*v*/*v*) peat/sand substrate under controlled environmental conditions (25 ± 1 °C with 16 h photoperiod).

### 2.2. Identification of PkWOX Family Genes in Korean Pine and Analysis of Physicochemical Properties

To identify WOX proteins in Korean pine, the file CBFD_NFYB_HMF (PF00046) was downloaded from the Pfam database (https://pfam.xfam.org/, accessed on 20 April 2023) according to the Korean pine protein database and the Hidden Markov Model (HMM). The hmmsearch command in the HMMER (v3.1) software [29] was used to search the Korean pine protein database with the file PF00046.hmm and the identified sequences were integrated. After screening and identification, 21 WOX amino acid sequences of Korean pine were obtained. The properties of these amino acid sequences, including predicted molecular weight, isoelectric point, number of amino acids, aliphatic index, and GRAVY (Grand Average of Hydropathicity) score, were analyzed using the online program ExPASy (https://web.expasy.org/protparam/, accessed on 25 September 2024) [30].

### 2.3. Chromosomal Localization, Collinearity Analysis, and Synteny Analysis

The location information of *WOX* family genes was extracted from the gff file of Korean pine. The chromosomal localization was visualized in TBtools v2.003. Then, we use TBtools to calculate the Ka and Ks values. To predict the function of *WOX* family genes, the syntenic relationship of *WOX* genes between Korean pine, *P. tabuliformis,* and *A. thaliana* was explored by using MCScanX program with default parameters. The *WOX* genes replications of Korean pine, *A. thaliana,* and *P. tabuliformis WOX* genes were visualized in TBtools.

### 2.4. Multiple Sequence Alignment and Phylogenetic Analysis

The WOX amino acid sequences were aligned using MUSCLE [31] and checked for the presence of the conserved HB site. A maximum likelihood method was used to generate phylogenetic trees (1000 bootstrap replicates) using MEGA v7.0 [32]. The phylogenetic tree was polished using Itol v7.

### 2.5. Analyzing the Exon/Intron Structure and Conserved Motifs

Conserved protein motifs in PkWOXs were identified via MEME Suite (motif number = 11, default parameters) [33]. Gene structures were subsequently extracted and visualized using TBTools v2.210.

### 2.6. Cis-Element Analysis of the PkWOX Promoters

The 2-kb genomic sequences upstream of the transcription start site were designated as promoter regions for all *PkWOX* family members. Putative cis-regulatory elements within these regions were predicted using PlantCARE (http://bioinformatics.psb.ugent.be/webtools/plantcare/html/, accessed on 8 November 2024) and subsequently visualized through TBtools v2.210 [34].

### 2.7. GO Enrichment Analysis of the PkWOX Genes

TBtools v2.210 was used to visualize the enrichment of GO entries of *PkWOXs*.

### 2.8. Tissue-Specific Expression Patterns of the PkWOX Genes

Transcriptomic analysis was performed to examine *PkWOX* genes expression patterns across multiple Korean pine tissues, including vegetative organs (roots, stems, leaves), reproductive tissues (seeds), in vitro cultures (embryogenic and non-embryogenic calli), and embryonic materials (somatic and zygotic embryos). Heatmap visualization of tissue-specific expression patterns was generated using TBtools v2.210 to compare *PkWOX* transcript levels across different sample types.

### 2.9. Expression Patterns of PkWOX Genes Under Abiotic Stress

Based on transcriptomic data from Korean pine embryogenic callus under abiotic stress treatments (abscisic acid and drought) in our study, we analyzed the expression profiles of *PkWOX* genes. Heatmap visualization of tissue-specific expression patterns was generated using TBtools v2.210 to compare *PkWOX* transcript levels across different sample types.

### 2.10. Cloning of the PkWOX Genes of Korean Pine

Full-length *PkWOX* cDNAs were PCR-amplified (Bioer, Hangzhou, China) [35] with specific primers (Appendix A), cloned into DH5α *E. coli* (ANGYUBIO, Shanghai, China), and verified by sequencing.

### 2.11. Subcellular Localization of PkWOXs

The coding sequences of the full-length *PkWOX* genes were PCR-amplified using gene-specific primers (Appendix A) and subsequently cloned into the *pFGC-eGFP* expression vector. This created C-terminal GFP fusion constructs under the control of the CaMV 35S promoter. For localization studies, two constructs—the empty *pFGC-eGFP* vector (control) and *PkWOX-eGFP* fusion vectors—were transiently expressed in tobacco epidermal cells via biolistic transformation. Subcellular localization was assessed by confocal microscopy (Zeiss LSM800, Shanghai, China) at 48 h post-transformation.

### 2.12. Yeast-Based Transcriptional Activity Assay

To evaluate the transcriptional regulatory potential of PkWOX proteins, we performed GAL4-based reporter assays in yeast. The coding sequences of *PkWOX* genes were PCR-amplified (primers in Appendix A) and directionally cloned into the *BamHI* site of pGBKT7 (Clontech), creating C-terminal fusions with the GAL4 DNA-binding domain.

Yeast strain Y2H Gold was co-transformed with

(1)pGBKT7-PkWOX constructs;(2)Empty pGBKT7 vector (negative control).

Transformants were selected on

-SD/-Trp (Coolaber, Beijing, China) (transformation control);-SD/-Trp/-His/-Ade (Coolaber, Beijing, China) (activation stringency test);-X-α-Gal (WEIDI, Shanghai, China) plates (reporter activation assay).

Yeast cultures were grown at 30 °C for 3–5 days before activation analysis.

### 2.13. Data Statistics and Analysis

The gene expression data were analyzed using one-way analysis of variance (ANOVA) followed by Duncan’s multiple range test in IBM SPSS Statistics for Windows, Version 27.0 (IBM Corp., Armonk, NY, USA) to determine significant differences among groups. A *p*-value < 0.05 was considered statistically significant.

## 3. Results

### 3.1. Genome-Wide Characterization and Physicochemical Profiling of the WOX Gene Family in Korean Pine

Based on the conserved domain CBFD_WOX_HMF (PF00046) specific to *WOX* family members in plants, 21 members of the *WOX* family of Korean pine were identified and named *PkWOX1*-*21* according to their position on chromosomes I to XI.

Comprehensive molecular characteristics of the 21 identified PkWOX transcription factors are summarized in Table 1, including genomic localization (locus identifiers and chromosomal positions), protein features (amino acid length, predicted molecular mass), physicochemical properties (theoretical pI values), and structural parameters (aliphatic index and GRAVY hydrophobicity scores)

The putative PkWOX protein sequences contained 164 to 488 aa, with molecular weights between 18,456.30 and 53,893.18 Da and isoelectric point values between 5.56 and 9.80. According to the predicted results, the average hydrophilicity coefficient of the PkWOXs protein is between −0.993 and −0.565. All were classified as hydrophilic proteins (Table 1).

### 3.2. Phylogenetic Analysis of Multiple Species, Protein Sequence Alignment, and Conservative Motif Analysis of PkWOXs

The evolutionary relationships among Korean pine WOX proteins were investigated through multiple sequence alignment and phylogenetic reconstruction (Figure 1). Comparative analysis of gene structures, including exon-intron organization, further elucidated the functional diversification within the *PkWOX* gene family. A maximum likelihood phylogenetic tree was constructed using MEGA v7.0 software to analyze the phylogenetic relationships between the members of the *WOX* gene families in multiple species. According to the classification of *A. thalian*, *PkWOX1*, *PkWOX10*, *PkWOX11*, *PkWOX12*, *PkWOX13*, *PkWOX14*, *PkWOX15*, *PkWOX17*, *PkWOX18*, and *PkWOX21* belonged to the modern clade (MC), *PkWOX2*, *PkWOX3*, *PkWOX4*, *PkWOX5*, *PkWOX6*, *PkWOX7*, *PkWOX8,* and *PkWOX9* belonged to the intermediate clade (IC), and *PkWOX16*, *PkWOX19,* and *PkWOX20* belonged to the ancient clade (AC) (Figure 1a). All members of the Korean pine PkWOX transcription factor contain conserved homologous and heterotypic domains. The conserved homologous heterotypic domain contains 60–66 amino acid residues characterized by a “helix-loop-helix-turn-helix” (Figure 1b). Only nine protein sequences of the *PkWOX* gene family contain WUS box motifs (Figure 1c).

Protein motif conservation in *PkWOX* family members was determined using MEME (https://meme-suite.org/meme/tools/meme, accessed on 14 March 2024) analysis, with results presented as a structural schematic (Figure 2a). The motif sequences of Pinus koraiensis WOX genes are provided in Appendix A. The results showed that the tissue form of the conserved motifs of WOX transcription factors within the same subfamily showed high consistency, indicating that *WOX* members within the same subfamily may undertake similar biological functions, which indirectly reflects the reliability of phylogenetic analysis. It is worth noting that all members of the WOX transcription factor family contain Motif 1 and Motif 4. Further analysis indicates that Motif 1 and Motif 4 form homologous and heterotypic domains. Meanwhile, there are conserved motifs with specific distributions within different subfamilies.

Analysis of exon–intron structures in Korean pine *WOX* genes (Figure 2b) provided valuable evolutionary information. Notably, *PkWOX16*, *PkWOX19,* and *PkWOX20* were found to contain exceptionally long introns, suggesting potential functional specialization. The consistent correlation between gene structures, conserved motif arrangements, and phylogenetic clustering strongly supports the reliability of the current classification system for these transcription factors. This integrated structural and evolutionary analysis offers important insights into the diversification of the *PkWOX* gene family in conifers.

### 3.3. Chromosomal Location, Collinearity Analysis, and Ka/Ks Calculation

Chromosomal mapping revealed an uneven distribution of *PkWOX* genes across the Korean pine genome. Among the 12 chromosomes, 21 identified *PkWOX* members were localized to seven specific chromosomes: Chr02, Chr04, Chr05, Chr09, Chr10, Chr11, and Chr12 (Figure 3).

In order to further analyze the evolutionary relationship of the *PkWOX* family, McScan X was used to perform collinearity analysis on the *A. thaliana* genome, Korean pine genome, and *P. tabulaeformis* genome under default parameters (Figure 4). The results indicated that there is no collinearity between the *WOX* gene of Korean pine and *A. thaliana*. The results showed that the Pkor02G02173.1 and the Pt5G38250.1, the Pkor02G02661.1 and the Pt5G39360.1, the Pkor02G02159.1 and the Pt5G37140.1, the Pkor04G04148.1 and the Pt6G50340.1, the Pkor05G01580.1 and the Pt8G00270.1, the Pkor11G03014.1 and the PtJG03550.6, and the Pkor12G02237.1 and the PtQG16860.1 were all located in the collinear blocks of the genomes of Korean pine and *P. tabulaeformis*, indicating a collinear relationship. This further suggests that the identified *PkWOX* gene in Korean pine is directly homologous to the *PtWOX* gene in *P. tabulaeformis*.

To assess selective pressures acting on the *PkWOX* gene family, we computed the nonsynonymous-to-synonymous substitution rate ratios (Ka/Ks) for paralogous gene pairs (Appendix A). A sequence pair with Ka/Ks > 1 implies positive or Darwinian selection; Ka/Ks = 1 means that both sequences drift neutrally; and Ka/Ks < 1 implies purifying selection. Selection pressure analysis revealed that the majority of duplicated *PkWOX* gene pairs underwent purifying selection (Ka/Ks < 1), indicating evolutionary constraints on protein-coding sequences.

### 3.4. Analysis of the Cis-Elements of the PkWOX Promoter

Analysis of *PkWOX* promoter regions (Figure 5; Appendix A) revealed a complex regulatory landscape characterized by enrichment of diverse cis-elements. The promoters contained abundant hormone-responsive elements, including motifs for auxin, abscisic acid, gibberellin, methyl jasmonate, and salicylic acid signaling. Numerous stress-responsive elements were also identified, particularly those associated with drought, low-temperature adaptation, and defense responses. Notably, light-responsive elements represented the most prevalent category, suggesting strong photoregulation of *PkWOX* expression. Additionally, tissue-specific regulatory elements related to meristem development and endosperm expression were detected. These findings collectively indicate that *PkWOX* genes likely function as key integrators of hormonal signals, environmental stresses, and developmental cues in Korean pine, providing new insights for future investigations into their regulatory roles in conifer biology.

### 3.5. GO Enrichment Analysis of PkWOX

Gene Ontology (GO) enrichment analysis was performed to investigate the biological roles of *PkWOX* genes (Figure 6). The results demonstrated significant enrichment in biological processes including embryonic development, reproductive system formation, and post-embryonic growth, consistent with the known functions of *WOX* family members. At the molecular level, PkWOX proteins exhibited DNA-binding transcription factor activity and transcriptional regulation capacity. These findings not only confirm the involvement of *PkWOX* genes in critical developmental processes but also suggest their potential roles in mediating plant responses to various biotic and abiotic stress factors, highlighting their importance as transcriptional regulators in Korean pine.

### 3.6. Tissue Expression Specificity and Abiotic Stress Expression Pattern

Analysis of *PkWOX* gene expression across multiple tissue types (Figure 7) revealed distinct spatial patterns among family members. While *PkWOX16* demonstrated ubiquitous expression in all examined tissues including embryonic callus, non-embryonic callus, somatic embryos, zygotic embryos, seeds, bark, phloem, cambium, xylem, roots, leaves, and branches, *PkWOX2* and *PkWOX3* exhibited highly specific expression profiles. These two genes showed elevated transcript levels exclusively in embryogenic tissues (embryonic callus, non-embryonic callus, somatic embryos, and zygotic embryos), with negligible expression detected in vegetative organs. This differential expression pattern suggests specialized functional roles for specific *PkWOX* members during embryogenesis versus broader regulatory functions for *PkWOX16* across various developmental contexts. Statistical analysis was conducted using one-way analysis of variance (ANOVA) with Duncan’s multiple range test in SPSS to compare the expression levels of individual genes among different tissue regions (Appendix A).

As shown in Figure 8a, the expression patterns of *PkWOX* members were visualized under different concentrations of gellan gum. *PWOX2*, *3,* and *16* were strongly expressed in all treatments and at all stages. Others were almost not expressed in all treatments and stages. Compared to the control, *PkWOX2*, *3,* and *16* showed no significant fluctuation in expression in all gellan gum treatments. Interestingly, based on the above analysis, we only analyzed the expression levels of three genes, *PkWOX2*, *3,* and *16*, under ABA treatment and found that *PkWOX2* and *16* were induced and upregulated by ABA; however, *PkWOX3* was induced and downregulated by ABA (Figure 8b). Statistical analysis was conducted using SPSS v27, where one-way ANOVA, combined with Duncan’s multiple range test, was applied to examine differential gene expression across different abiotic stress conditions (Appendix A).

### 3.7. Subcellular Localization and Transcriptional Activity Assessment of Transcription Factors

Based on their distinct expression profiles, PkWOX2, PkWOX3, and PkWOX16 were selected for further functional characterization. Subcellular localization analysis revealed that all three proteins were detected in both the nucleus and cell membrane (Figure 9), exhibiting a distribution pattern comparable to the empty vector control. This dual localization suggests potential roles in membrane-associated signaling and nuclear transcriptional regulation, consistent with known functions of *WOX* family transcription factors. The nuclear presence particularly supports their predicted role in gene regulation, while membrane localization may indicate additional non-canonical functions in cellular signaling pathways. These findings provide important insights into the cellular behavior of these PkWOX proteins and their potential mechanisms of action in Korean pine.

The transcriptional activation potential of *PkWOX* proteins was evaluated using the GAL4-based yeast two-hybrid system (Figure 10). Yeast strains expressing *PkWOX*-GAL4 DNA-binding domain (GAL4DB) fusions showed normal growth on SD/-Trp selection medium, confirming successful transformation. However, on stringent selection media (SD/-Trp/-His/-Ade containing X-α-gal), differential growth patterns were observed: While PkWOX16 transformants exhibited growth characteristics identical to the negative control (indicating absence of autoactivation activity), PkWOX2 and PkWOX3 demonstrated robust growth, clearly establishing their intrinsic transcriptional activation capability. These results reveal functional divergence among PkWOX family members, with PkWOX2 and PkWOX3 possessing autonomous transactivation domains that can drive reporter gene expression in this heterologous system.

### 3.8. Prediction of Protein Interaction Networks for PkWOX Genes

For *PkWOX2*, *3*, *17,* and *19* predicted 15, 1, 1, and 2 interacting proteins (Figure 11). A description of the interacting proteins can be found in Appendix A. Line width indicates predicted PPI strength.

## 4. Discussion

### 4.1. Roles of WOX Transcription Factors in Plant Growth and Morphogenesis

The *WOX* gene is involved in processes such as stem cell differentiation, leaf primordia formation, and in vitro plant regeneration [14], and plays an important role in plant growth and development. Conducting research on the function of *WOX* genes is of great significance for deciphering the regulatory network of plant development.

Our comprehensive analysis identified 21 *WOX* family genes in the Korean pine genome. These transcription factors exhibited predominant expression in the shoot apical meristem (SAM), root apical meristem (RAM), and vascular cambium, where they function as key regulators of meristematic activity. The *PkWOX* genes appear to play dual roles in maintaining stem cell populations—both stimulating proliferative cell division and suppressing premature cellular differentiation, consistent with the conserved functions of *WOX* genes in other plant species [4].

Phylogenetic and expression analyses revealed distinct functional specialization among *PkWOX* gene family members. *PkWOX16*, representing the ancient clade, exhibited constitutive high-level expression across all examined tissues (Figure 7 and Figure 8), indicative of potential housekeeping roles in fundamental cellular processes. In contrast, intermediate clade members demonstrated pronounced tissue specificity, with *PkWOX2* and *PkWOX3* showing particularly strong preferential expression in embryogenic tissues including embryonic callus, non-embryogenic callus, somatic embryos, and zygotic embryos. These expression patterns correlate with the established functions of *WOX* family genes in regulating callus formation and maintenance, suggesting specialized roles for intermediate clade members in embryogenic processes while ancient clade members may maintain more general cellular functions in Korean pine (Figure 7). In *A. thaliana*, the transcription factor LBD16 is specifically expressed in CIM (callus-inducing medium) after being activated by *WOX11*. The *WOX11*-*LBD16* pathway is a guarantee that callus tissue acquires pluripotency [36]. The *OsWOX11* transcript is present in rice callus tissue, suggesting that *OsWOX11* can affect the growth of rice callus tissue [37]. This suggests that the *WOX* subfamily may be involved in callus tissue development.

Phylogenetic conservation of expression patterns was observed between WUS-clade members in Korean pine and their *A. thaliana* orthologs. Notably, *PkWOX1* exhibited pronounced tissue-specific expression, with high transcript accumulation in vascular tissues (phloem, cambium, and xylem) and roots, while showing minimal expression in other examined organs (Figure 1 and Figure 7). This spatial expression profile mirrors the known functions of WUS-clade genes in vascular development and root growth, suggesting evolutionary conservation of regulatory mechanisms between gymnosperms and angiosperms. Its counterpart *AtWOX4* is present throughout the life cycle of plant vascular tissues and is mainly responsible for promoting and maintaining the dynamic balance of cell numbers between xylem and phloem and participating in the regulation of vascular differentiation in the vascular cambium. It is also negatively regulated by the *CLE* gene *FCP1* [38,39,40].

*PkWOX2* and *PkWOX3* exhibited pronounced expression specificity, with elevated transcript levels exclusively detected in embryogenic tissues (embryonic callus, non-embryogenic callus, somatic embryos, and zygotic embryos). Their expression profiles followed a developmental trajectory, showing progressive downregulation during embryo-to-seedling transition until becoming undetectable in mature tissues. This spatiotemporal pattern suggests their specialized roles in early developmental programming of *Pinus koraiensis*, potentially serving as molecular markers for embryogenic competence.

These observations align with conserved functions of *WOX* orthologs across plant species: *AtWOX5* displays root quiescent center-specific expression [41], *PaWOX2* shows peak expression during early embryogenesis [16], and *PmWOX2*/*3*/*4* exhibit callus-preferential expression with minimal detection in seeds or seedlings. Such evolutionary conservation underscores the crucial, stage-specific regulatory roles of particular *WOX* family members during plant embryogenesis and meristem maintenance.

### 4.2. Proteins Exhibit Canonical Transcription Factor Characteristics

WOX transcription factors contain four key domains: DNA-binding homeodomain, regulatory region, NLS, and oligomerization site, as demonstrated by functional studies of *AtWUS*, *AtWOX3*/*4*/*11* in *A. thaliana* [31], which are localized in the nucleus in *A. thaliana*. *LkWOX4* in *Larix kaempferi* is a nuclear localization protein [42].

This study represents the first comprehensive functional characterization of WOX family proteins in Korean pine, examining both their transactivation potential and subcellular localization patterns. Our analysis revealed two functionally distinct classes among the examined *PkWOX* members: PkWOX2 and PkWOX3 demonstrated strong autonomous transcriptional activation capability and contained functional nuclear localization signals, exhibiting dual localization in both nuclear and cell membrane compartments, which suggests their capacity for direct transcriptional regulation of downstream targets. In contrast, PkWOX16 displayed similar nuclear-membrane partitioning but lacked intrinsic transactivation activity, implying a potential role as a transcriptional corepressor that may modulate gene expression through protein–protein interactions rather than direct DNA binding-mediated activation. These findings provide novel insights into the functional diversification of WOX transcription factors in gymnosperms and suggest distinct regulatory mechanisms operating during conifer development, with important implications for understanding the evolutionary conservation and divergence of this critical plant-specific transcription factor family.

### 4.3. Differential Expression Patterns of PkWOX Genes in Response to Environmental Stressors

Plants must adapt their growth and development to cope with environmental changes. Although most research on *WOX* genes has focused on developmental regulation [43], our study reveals their important roles in stress responses. The *PkWOX* genes in Korean pine show significant expression changes under environmental stresses, consistent with the stress-responsive characteristics reported for *WOX* genes in rice [44]. These findings expand our understanding of *WOX* gene functions beyond development to include environmental adaptation.

Figure 8 summarizes the dual regulatory roles of *PkWOX* family members in both organ development and environmental stress responses. Notably, the phytohormone abscisic acid (ABA) emerges as a key mediator of abiotic stress adaptation, potentially influencing PkWOX-mediated stress response pathways in Korean pine [45]. Under drought conditions, plants reduce water release mainly through stomatal closure, and ABA can effectively induce stomatal closure. *PkWOX2* had the maximum expression at 12 h, and *PkWOX3* at 6 h (Figure 8a). The expression of *PkWOX16* showed almost no change compared to the control group. The expression of *PkWOX2* was similar in drought and ABA treatment (Figure 8a).

Notably, *PkWOX3* expression was significantly downregulated by ABA treatment, whereas *PkWOX2* showed marked induction under the same conditions (Figure 8b). Promoter analysis of these genes identified the presence of ABA-responsive cis-regulatory elements (Figure 5), providing a molecular basis for their divergent transcriptional responses to this phytohormone. These opposing expression patterns suggest distinct functional roles for these *WOX* family members in ABA-mediated stress responses in Korean pine. This further supports the evolutionary conservation of these stress responses.

## 5. Conclusions

This study presents a comprehensive analysis of the WOX transcription factor family in Korean pine, identifying 21 distinct members through bioinformatic approaches. Phylogenetic classification revealed these genes cluster to be in three evolutionarily conserved clades. *PkWOX2*, *3,* and *16* were located in the nucleus and in the cell membrane. The *PkWOX2* and *3* proteins showed transcriptional self-activation activity, whereas *PkWOX4* did not. *PkWOX2*, *3*, *17,* and *19* predicted 15, 1, 1, and 2 interacting proteins. Expression profiling revealed strong correlations between phylogenetic relationships and tissue-specific expression patterns across different clades, from ancient to WUS-clade members. These findings establish fundamental knowledge about the *PkWOX* gene family and provide important insights for future investigations into their roles in somatic embryogenesis and other developmental processes in conifers.

## Figures and Tables

**Figure 1 biology-14-00411-f001:**
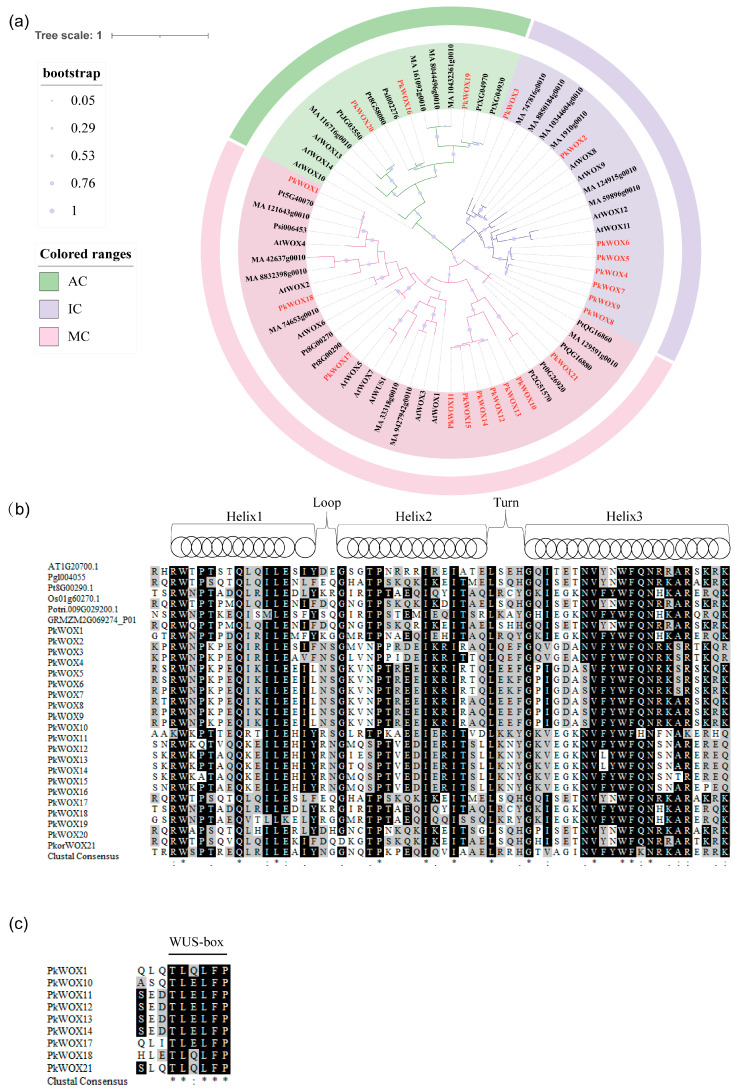
Phylogenetic analysis and multiple alignment of the amino acid sequences of 21 *PkWOX* proteins. (**a**) Cross-species phylogenetic analysis of WOX proteins, including *A. thalian*, Korean pine, *Pinus tabuliformis*, *P. abies,* and *Picea sitchensis.* The phylogenetic tree was constructed using the maximum likelihood method in MEGA, with 1000 bootstrap replicates under the Poisson model. The gene classes are indicated by different colors. The taxon names are shown in pink for the modern clade (MC), in purple for the intermediate clade (IC) and in green for the ancient clade (AC). The members of the *WOX* protein from Korean pine are shown in red. (**b**) Amino acid alignment of the conserved domains of *WOX* proteins from different organisms at, *A. thalian*; Pk, Korean pine; Pgl, *Picea glauca*; Pt, *P. tabuliformis*; Os, *O. sativa*; Potri, *P. trichocarpa*; GRMZM, *Z. mays*. (**c**) Alignment of the C-terminal region in a subgroup of the WUS family. In (**b**,**c**), identical residues are shown in black boxes, conservative changes in gray. Identity between residues is indicated by an asterisk, conservative changes are indicated by double dots, and less-conservative changes are indicated by single dots.

**Figure 2 biology-14-00411-f002:**
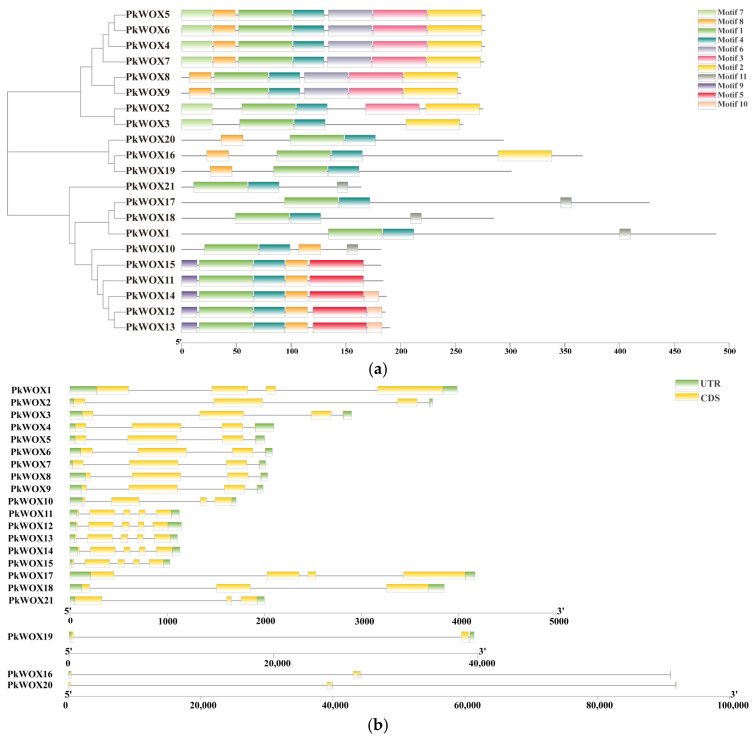
Structural characterization of PkWOX transcription factors in Korean pine. (**a**) Distribution of conserved protein motifs identified through predictive analysis, with distinct colors representing different motif types. (**b**) Genomic organization of *PkWOX* genes, depicting coding exons (yellow), non-coding flanking regions (green), and intronic sequences (black lines).

**Figure 3 biology-14-00411-f003:**
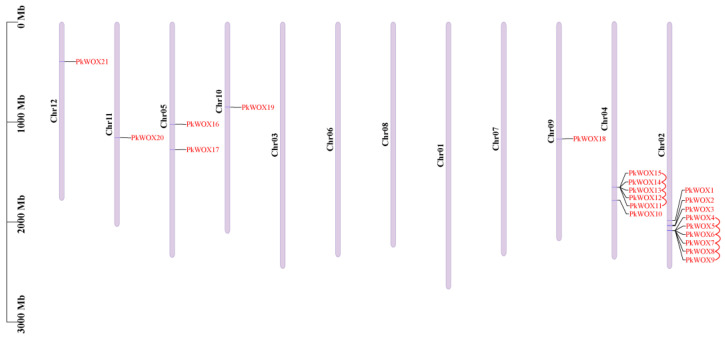
Genomic distribution and segmental duplications of *PkWOX* genes in Korean pine. Chromosomal positions are marked with scale bars (Mb) and duplicated gene pairs are connected by red lines. Gene labels and chromosome numbers are displayed in red and on the left, respectively.

**Figure 4 biology-14-00411-f004:**
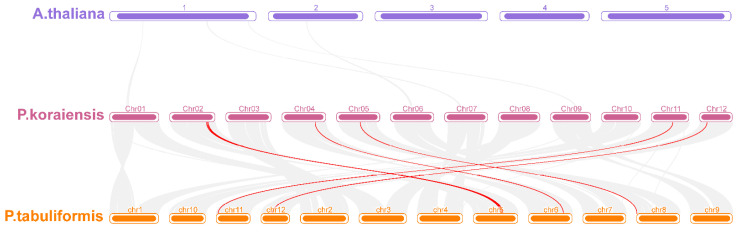
Collinearity analysis of *WOX* genes in Korean pine, *A. thaliana*, and *P. tabulaeformis*. 1–5 represents 5 chromosomes of *A. thaliana*; Chr01–Chr12 represents 12 chromosomes of *P. tabulaeformis*; and chr1–chr12 represents 12 chromosomes of Korean pine.

**Figure 5 biology-14-00411-f005:**
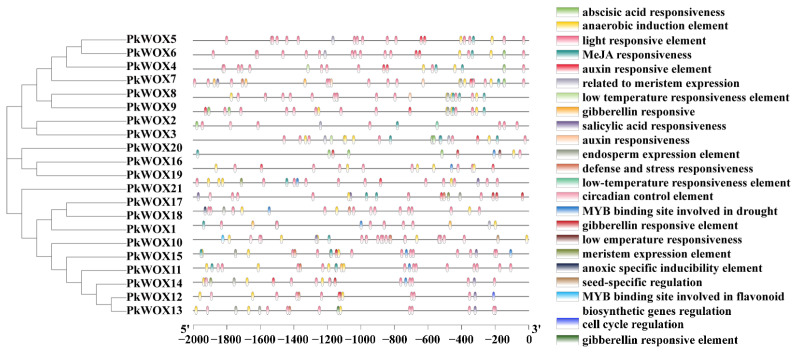
The 2-kb upstream sequences of *PkWOX* genes were analyzed for cis-regulatory elements using PlantCARE, followed by visualization with TBtools.

**Figure 6 biology-14-00411-f006:**
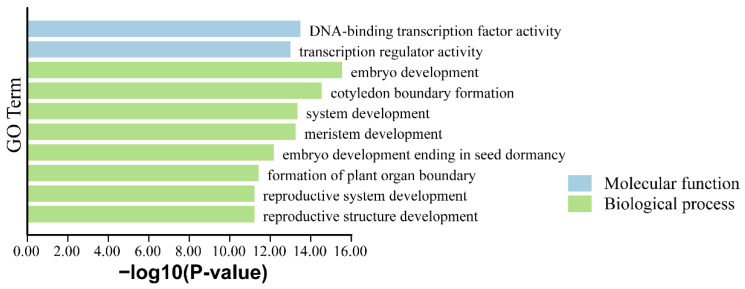
GO enrichment analysis of the members of the *PkWOX* family.

**Figure 7 biology-14-00411-f007:**
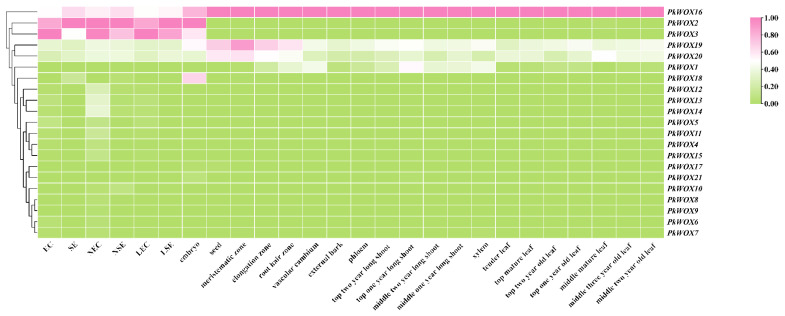
Spatiotemporal expression patterns of PkWOX genes displayed by heatmap analysis, with color gradient (pink = high, green = low) representing log2^(FPKM)^ values normalized to 0–1 scale. Tissues examined include embryogenic (EC) and non-embryogenic callus (NEC), somatic (SE) and zygotic embryos, seeds, and vegetative organs (roots, stems, leaves), with additional samples showing lost embryogenic capacity (LEC, LSE).

**Figure 8 biology-14-00411-f008:**
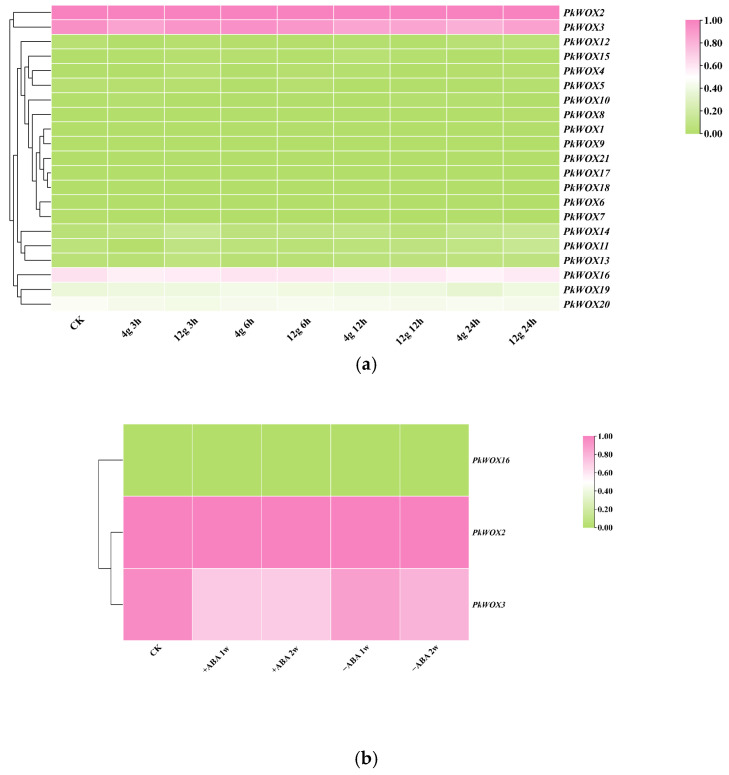
Expression analysis of *PkWOX* genes under ABA and different concentrations of gellan gum treatments. (**a**) Somatic embryos cultured on maturation media containing different gellan gum concentrations (4 g/L and 12 g/L) were compared with control samples (CK). (**b**) ABA treatment effects were evaluated by comparing somatic embryos grown on media supplemented with 80 µmol/L ABA (+ABA) versus ABA-free media (−ABA). Transcriptome data were normalized to control (CK) samples, with relative expression levels calculated as log2 ^[FPKM (treatment)/FPKM(CK)]^.

**Figure 9 biology-14-00411-f009:**
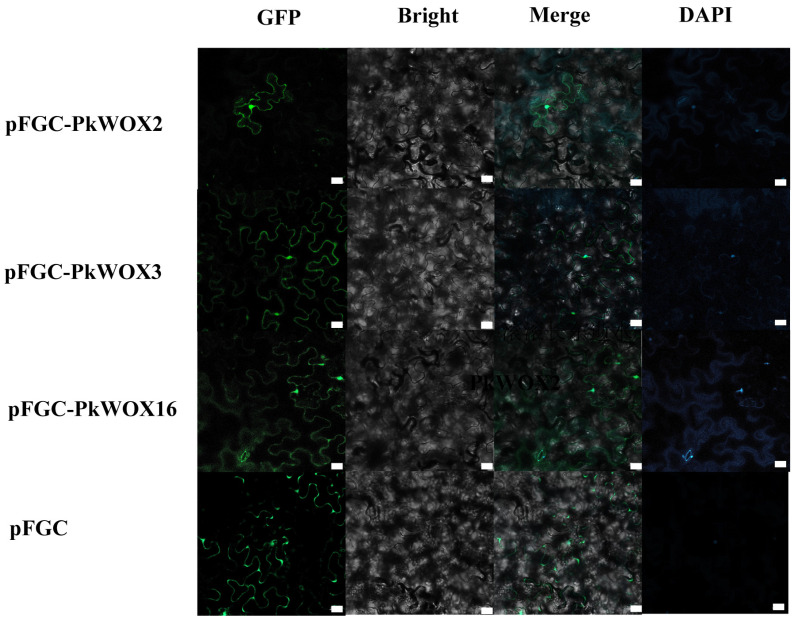
Subcellular localization of the *PkWOX* proteins. DAPI, a dye used to stain the nucleus; BF: bright field; Merge: Merged images of BF, GFP, and DAPI staining. White square: scale bar. Scale bar = 20 μm.

**Figure 10 biology-14-00411-f010:**
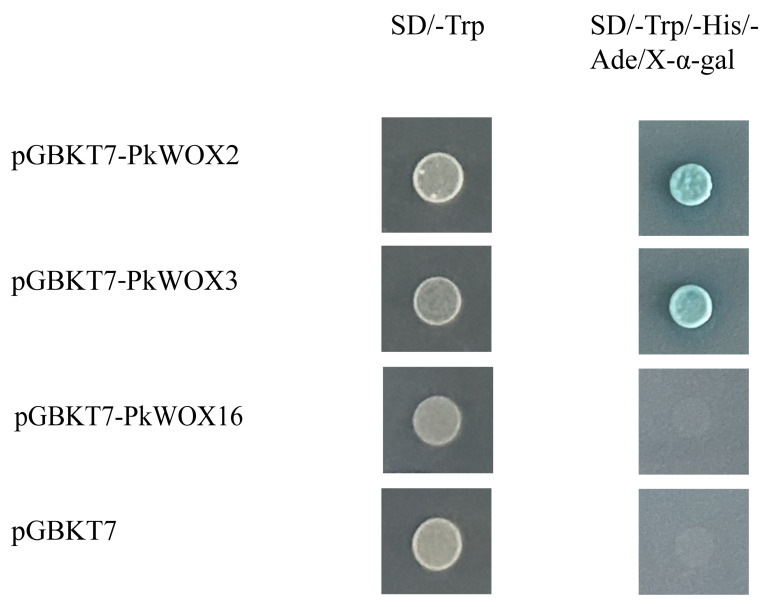
The transcriptional activation capability of *PkWOX* proteins was evaluated using a yeast-based assay system, with the pGBKT7 empty vector serving as the negative control.

**Figure 11 biology-14-00411-f011:**
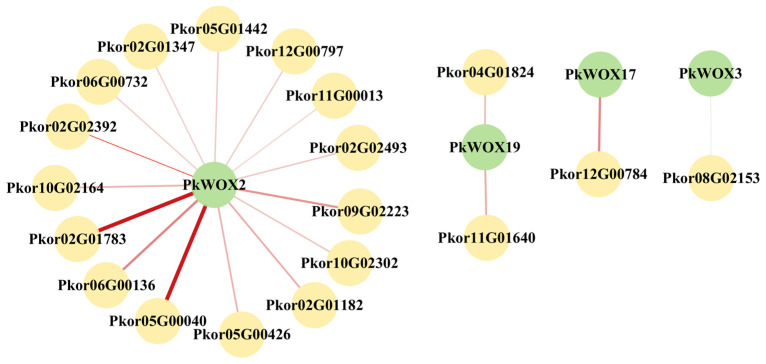
The protein interaction network of *PkWOX*. The green nodes represent the *PkWOX* proteins, while the yellow nodes represent the predicted interacting proteins. The depth of the connecting lines color indicates the strength of the interaction relationship.

**Table 1 biology-14-00411-t001:** Parameters for the 21 identified *PkWOXs* and deduced polypeptide sequences in the genome of Korean pine.

Gene Name	Locus Name	Amino Acid No.	Molecular Weight (Da)	Isoelectric Points	GRAVY	Chromosome Location
*PkWOX1*	Pkor02G02105	488	53893.18	6.61	−0.665	Chr02:1984412850..1984416411
*PkWOX2*	Pkor02G02159	275	30634.96	6.51	−0.682	Chr02:2032426825..2032430487
*PkWOX3*	Pkor02G02173	257	28472.43	6.06	−0.658	Chr02:2036932723..2036935410
*PkWOX4*	Pkor02G02661	277	30804.31	9.32	−0.752	Chr02:2084570897..2084572996
*PkWOX5*	Pkor02G02662	277	30806.36	9.56	−0.793	Chr02:2084651843..2084653845
*PkWOX6*	Pkor02G02663	277	30802.28	9.28	−0.791	Chr02:2084820524..2084822606
*PkWOX7*	Pkor02G02664	276	30820.43	9.47	−0.788	Chr02:2084926777..2084928789
*PkWOX8*	Pkor02G02665	255	28645.47	9.80	−0.565	Chr02:2085544487..2085546521
*PkWOX9*	Pkor02G02666	255	28670.47	9.72	−0.571	Chr02:2085656072..2085658059
*PkWOX10*	Pkor04G04148	182	21119.85	9.74	−0.903	Chr04:1787036381..1787038091
*PkWOX11*	Pkor04G04174	184	20798.41	9.52	−0.753	Chr04:1657982908..1657984034
*PkWOX12*	Pkor04G04175	186	20842.42	9.01	−0.744	Chr04:1657880486..1657881632
*PkWOX13*	Pkor04G04176	190	21186.68	9.15	−0.793	Chr04:1657819107..1657820213
*PkWOX14*	Pkor04G04177	187	21104.68	9.12	−0.775	Chr04:1657803316..1657804447
*PkWOX15*	Pkor04G04178	182	20565.93	7.73	−0.782	Chr04:1657787946..1657788974
*PkWOX16*	Pkor05G01247	360	40306.97	5.96	−0.790	Chr05:1020056105..1020146943
*PkWOX17*	Pkor05G01580	427	48231.86	7.65	−0.761	Chr05:1275855869..1275859721
*PkWOX18*	Pkor09G01265	285	32205.45	5.56	−0.836	Chr09:1166586641..1166590206
*PkWOX19*	Pkor10G00929	301	34200.94	7.21	−0.993	Chr10:852352721..852392372
*PkWOX20*	Pkor11G03014	294	32964.98	5.57	−0.601	Chr11:1155597034..1155688998
*PkWOX21*	Pkor12G02237	164	18456.30	5.83	−0.729	Chr12:395821012..395823012

## Data Availability

The datasets supporting the conclusions of this article are included within the article and its Appendix A.

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
