# Peer review of "Genome-Wide Identification of WOX Genes in Korean Pine and Analysis of Expression Patterns and Properties of Transcription Factors"

_biology, 2025, doi:10.3390/biology14040411_

Round 1
Reviewer 1 Report
Comments and Suggestions for Authors
The study of the WOX gene family in Korean pine yielded several important results. A total of 21 members of the PKwox gene family were identified using bioinformatics methods using the entire Korean pine genome database. This comprehensive analysis provides understanding the roles of these genes in various biological processes. The 21 PKwox genes were unevenly distributed across 7 of the 12 chromosomes in the Korean pine genome, highlighting the complexity of their genomic organization. The promoters of the PKwox genes contained various cis-acting elements required for gene regulation, including those responding to stress, light, hormones, and meristem regulation. The results suggest that foundt genes may play important roles in response to environmental conditions. The authors revealed that PKwox16 was expressed in all tissues, while PKWOx2 and PKwox3 was expressed in embryonic callus, non-embryonic callus, somatic embryos and zygotic embryos. This indicates their potential importance in the embryogenesis of Korean pine. These results provide fundamental insights into the WOX gene family of Korean pine and open the way for further studies on their role in somatic embryogenesis and other developmental processes. The text of the article is very well written. The results of the bioinformatic analysis are illustrated in details. As a minor remark, in Figure 11 the colors in the figure and in the legend do not correspond to each other.
Author Response
Dear Reviewer,
We are very grateful to you for taking the time to read and modify our article again. We find that your comments play a very important role in improving the quality of our papers. We have carefully revised the paper in light of your comments, and please find our response to the comments made below. We marked the modified part of the manuscript in green.
The results of the bioinformatic analysis are illustrated in details. As a minor remark, in Figure 11 the colors in the figure and in the legend do not correspond to each other.
Thank you for the reviewer's comments. I have meticulously remade Figure 11, annotated its legend with precision, and cross-checked to ensure the colors in the figure and the legend are in perfect concordance(Line 421-422).
Reviewer 2 Report
Comments and Suggestions for Authors
Review report on the manuscript ID “3527990”, entitled “Genome-wide identification of WOX genes in Korean pine and analysis of expression patterns and properties of transcription factors”. The manuscript presents a comprehensive genome-wide analysis of WOX genes in Korean pine, identifying 21 members and analyzing their phylogenetic relationships, expression patterns, and transcriptional activities. The study provides valuable insights into the functional roles of WOX genes in plant development, particularly in somatic embryogenesis. The use of bioinformatics and molecular biology techniques strengthens the study. However, there are areas that need improvement in terms of clarity, presentation, and discussion depth.
Major comments:
- In the introduction, some references are outdated; consider incorporating more recent studies.
- In the methods, the transcriptome dataset used for expression analysis is unpublished. If possible, provide additional details on its quality and the normalization methods used.
- In the methods, the statistical analyses applied to the gene expression data are not described and should be included.
- In the results, the figure legends lack clarity, particularly for the phylogenetic trees and expression heatmaps; more detailed explanations are needed.
- In the results, the statistical interpretations are underdeveloped; for instance, significance testing in the gene expression data should be addressed.
- The discussion on protein-protein interaction networks is brief and should be elaborated upon.
- How do these findings compare to previous studies on other conifers or closely related species?
Comments on the Quality of English Language
Minor comments:
Line 13: "PkWOXs protein" should be "PkWOX proteins"
Line 50: "have been gradually been elucidated" is redundant. It should be "have gradually been elucidated"
Line 63: Norwegian spruce (Picea abies)
Line 80: hormonal, biotic, and abiotic stress
Line 82: Korean pine is a constructive species of the deciduous forest,
Line 84: "high-value and valuable" is redundant. It should be "high-value"
Author Response
Dear Reviewer,
We are very grateful to you for taking the time to read and modify our article again. We find that your comments play a very important role in improving the quality of our papers. We have carefully revised the paper in light of your comments, and please find our response to the comments made below. We marked the modified part of the manuscript in yellow.
Major comments:
Introduction:
- In the introduction, some references are outdated; consider incorporating more recent studies.
Thank you for the reviewer's comments. Regarding the issue you raised about the earlier references cited, we fully understand your concern. These early references are included because they represent the seminal research work in which the gene family (such as the WOX gene family) was first discovered and characterized. These studies laid the foundation for the field, and much of the subsequent research has been built upon them. Therefore, although these references were published some time ago, their pioneering significance in the field cannot be overlooked.
At the same time, we highly value your suggestion regarding the timeliness of the references. In the revised manuscript, we have supplemented the discussion with recent advancements related to this gene family to reflect the latest developments and trends in the field(lines85-88). Additionally, we have provided a clearer explanation of the citations of the early references to highlight their historical importance and scientific value (Lines 52-53).
Once again, we sincerely appreciate your valuable feedback, which has greatly improved our manuscript. If you have any further suggestions regarding the revised content, we would be more than happy to make additional adjustments.
Methods:
- In the methods, the transcriptome dataset used for expression analysis is unpublished. If possible, provide additional details on its quality and the normalization methods used.
We sincerely appreciate the reviewer's valuable comment.
Data Filtering:
The raw sequencing data were filtered using SOAPnuke (v1.5.2) [1] to remove:
- reads containing adapters (adapter contamination);
- reads with unknown base (N) content exceeding 10%;
- low-quality reads (defined as reads where bases with a quality score below 15 account for more than 50% of the total bases). The resulting high-quality data were designated as clean data.
Reference Genome Alignment:
The clean data were aligned to the reference genome sequence using HISAT2 (v2.1.0) [2].
Reference Gene Alignment:
The clean data were aligned to the reference gene sequences using Bowtie2 (v2.2.5) [3], followed by gene expression quantification for each sample using RSEM (v1.2.8) [4].
Gene Annotation:
Known genes were annotated against seven major functional databases (KEGG, GO, NR, NT, SwissProt, Pfam, and KOG), and transcription factor prediction was performed.
Time-Series and WGCNA Analysis:
Time-series analysis was conducted using Mfuzz (v2.34.0) [5], and gene co-expression network analysis was performed using WGCNA (v1.48).
Differentially Expressed Genes (DEGs):
- Intra-group differential gene analysis was performed using DESeq with thresholds of |Fold Change| ≥ 2 and adjusted p-value ≤ 0.001. Inter-group comparisons were conducted using PossionDis with |Fold Change| ≥ 2 and FDR ≤ 0.001.
- Differentially expressed genes (DEGs) were visualized using the pheatmap function to generate clustered heatmaps.
- Functionally categorized differentially expressed genes (DEGs)[6] were analyzed based on GO and KEGG annotation results following official classification systems. Enrichment analyses were performed using:
(1) KEGG pathway analysis implemented via the phyper function in R software;
(2) GO term enrichment analysis using the GO::TermFinder package (https://metacpan.org/pod/GO::TermFinder).
A significance threshold of Q-value ≤ 0.05 was applied, with terms meeting this criterion considered significantly enriched among candidate genes.
References:
[1] Yuxin Chen, et al. SOAPnuke: a MapReduce acceleration-supported software forintegrated quality control and preprocessing of high-throughput sequencingdata.Gigascience. 2018 Jan 1;7(1):1-6
[2] Kim, D., Langmead, B. & Salzberg, S. L. HISAT: a fast spliced aligner with lowmemory requirements. Nat. Methods 12, 357-360 (2015).
[3] Langmead, B. et al. Fast gapped-read alignment with Bowtie 2. Nat. Methods 9,357-359 (2012).
[4] Li, B. & Dewey, C. N. RSEM: accurate transcript quantification from RNA-Seqdata with or without a reference genome. BMC Bioinformatics 12, 323 (2011).
[5] Kumar, L. & Futschik, M. E. Mfuzz: a software package for soft clustering ofmicroarray data. Bioinformation 2, 5-7 (2007).
If I have misinterpreted anything, I sincerely apologize and genuinely welcome your valuable feedback and corrections.
- In the methods, the statistical analyses applied to the gene expression data are not described and should be included.
We sincerely appreciate the reviewer's valuable comment. As suggested, we have now added detailed statistical methods for gene expression analysis in the Methods section (Lines 210-214).
Results:
- In the results, the figure legends lack clarity, particularly for the phylogenetic trees and expression heatmaps; more detailed explanations are needed.
We sincerely appreciate your constructive comments. We have comprehensively revised the figures in the Results section and modified their legends as follows:
- Phylogenetic trees:
The phylogenetic tree was constructed using the maximum likelihood method in MEGA6.06, with 1000 bootstrap replicates under the Poisson model (Lines 266-272).
- Expression heatmaps:
Supplemented the data processing method (Lines 360-364, 384-385).
- In the results, the statistical interpretations are underdeveloped; for instance, significance testing in the gene expression data should be addressed.
We sincerely appreciate your constructive comments. In response to the reviewer’s suggestion, additional results from the SPSS analysis are included in Supplementary Tables 5, 6, and 7 (Lines 355-368, 372-375).
Discussion:
- The discussion on protein-protein interaction networks is brief and should be elaborated upon.
We are grateful for the valuable comment regarding the protein-protein interaction network analysis. As suggested, we have regenerated the network diagrams using Cytoscape, where the thickness of connecting edges represents the predicted interaction strength between protein pairs (Lines 416-417, 420-421). However, we acknowledge that this analysis is currently limited to predictive results without experimental validation. This represents a constraint of our present study. In subsequent work, we plan to conduct targeted experiments to verify these interactions and perform functional characterization of the identified proteins, which would substantially strengthen the biological significance of our findings.
- How do these findings compare to previous studies on other conifers or closely related species?
We sincerely appreciate the reviewer's valuable question regarding the comparison of our findings with previous studies on other conifers or closely related species. Based on the reviewer's suggestion, we have revisited the literature and reanalyzed our transcriptomic data to provide a more comprehensive discussion. Below, we summarize the key comparisons and insights:
- Summary of Our Findings:
Through re-analysis of the data and literature review, we found that PkWOX2 and PkWOX 3 may serve as a key biomarker for detecting embryonic development in this study.
- Comparison with Previous Studies:
Our findings are consistent with several studies on other conifers. For example, during embryogenesis, PaWOX2 exhibits the highest expression levels in the early developmental stages but shows lower detection levels in seedling tissues. Additionally, In Pinus massoniana, except for PmWOX1, the three genes PmWOX2, PmWOX3, and PmWOX4 show elevated expression levels in callus, and all of them did not tend to be expressed in seeds and young seedlings. These findings provide further evidence for the conserved evolutionary mechanisms in conifer species (Lines 462-474).
- Implications of Findings:
We believe these additions significantly strengthen the manuscript and provide a clearer context for our findings within the broader field of conifer research.
- Future Directions:
Future studies should explore the validation of PkWOX2/3 as key biomarkers for embryonic development. Successful validation would significantly streamline the embryogenic callus induction process while improving the accuracy of embryogenic callus identification.
Minor comments:
Line 13: "PkWOXs protein" should be "PkWOX proteins"
Thank you for the reviewer's comments. Upon thorough scrutiny, I have corrected "PkWOXs protein" to "PkWOX proteins" (Line 14).
Line 50: "have been gradually been elucidated" is redundant. It should be "have gradually been elucidated"
Thank you for the reviewer's comments. Due to my revision of the article, the original line 50 has become line 51 now. I have removed "been" in accordance with your suggestion (Line 51).
Line 63: Norwegian spruce (Picea abies)
Thank you for the reviewer's comments. Due to my revision of the article, the original line 63 has become line 65 now. I am sincerely ashamed of my negligence. It has now been revised to "Norway spruce"(Lines 65).
Line 80: hormonal, biotic, and abiotic stress
Thank you for the reviewer's comments. Due to the redundancy in this section and its similarity in meaning to the first sentence of this paragraph, it has been removed.
Line 82: Korean pine is a constructive species of the deciduous forest,
Thank you for the reviewer's comments. Due to my revision of the article, the original line 82 has become line 89 now. After consulting the literature, I have revised the description of the tree species to which the Korean pine belongs and its economic value (Lines 89-90).
Line 84: "high-value and valuable" is redundant. It should be "high-value"
Thank you for the reviewer's comments. Due to my revision of the article, the original line 84 has become line 91 now. I have removed "and valuable" as per your request(Line 91).
Round 2
Reviewer 2 Report
Comments and Suggestions for Authors
Accept